# Integrating Reference Intervals into Chimpanzee Welfare Research

**DOI:** 10.3390/ani13040639

**Published:** 2023-02-12

**Authors:** Jessica C. Whitham, Katie Hall, Lisa K. Lauderdale, Jocelyn L. Bryant, Lance J. Miller

**Affiliations:** 1Chicago Zoological Society—Brookfield Zoo, 3300 Golf Road, Brookfield, IL 60513, USA; 2Sedgwick County Zoo, 5555 W Zoo Boulevard, Wichita, KS 67212, USA; 3Independent Researcher, La Grange Highlands, IL 60525, USA

**Keywords:** reference interval, immunoglobulin-A, glucocorticoids, animal welfare, behavioral diversity

## Abstract

**Simple Summary:**

In recent years, animal welfare scientists have focused on developing novel approaches for improving the quality of life of chimpanzees living in zoos, sanctuaries, and laboratories. To evaluate the emotional, physical, and mental states of individual chimpanzees, welfare researchers are encouraged to integrate indicators that can be monitored non-invasively. In this study, we analyzed data from 40 adult chimpanzees from 16 zoological facilities to generate reference intervals (i.e., ranges of values) for behavioral and physiological measures of welfare. We also examined whether these measures varied by age or sex. While we discovered sex or age differences for a handful of behaviors, most behaviors and the physiological measures (fecal glucocorticoid metabolites, fecal immunoglobulin-A) did not vary by sex or age. Moving forward, animal care professionals can make informed decisions regarding management and husbandry practices by comparing an individual chimpanzee’s values to the reference intervals reported in this study.

**Abstract:**

Animal welfare researchers are committed to developing novel approaches to enhance the quality of life of chimpanzees living in professional care. To systematically monitor physical, mental, and emotional states, welfare scientists highlight the importance of integrating non-invasive, animal-based welfare indicators. This study aimed to create species-specific reference intervals for behavioral measures and physiological biomarkers. Specifically, we analyzed data from 40 adult chimpanzees (22 females, 18 males) residing at 16 zoological facilities to generate reference intervals for behavioral states and events, behavioral diversity, fecal glucocorticoid metabolites (GCMs), and fecal immunoglobulin-A (IgA). Comparisons of sex and age using linear regression models revealed significant differences for several behaviors. The proportion of time spent engaged in mutual/multiple social grooming significantly decreased as individuals aged. Furthermore, males spent a higher proportion of time performing aggressive contact behaviors and displaying to other chimpanzees when compared to females. Males also performed sexual examination behaviors at a higher rate than females. Behavioral diversity, fecal GCM, and fecal IgA did not vary by sex or age. In the future, values for individual chimpanzees can be compared to the ranges reported here for particular age/sex classes. Ultimately, animal care professionals can utilize reference intervals to make evidence-based decisions regarding management practices and environmental conditions.

## 1. Introduction

In recent years, animal welfare scientists have promoted the development of novel approaches for enhancing the quality of life of individual animals living in professional care. Animal welfare is measured on a continuum from poor to good and incorporates the individual’s emotional, physical, and mental states [1]. To monitor and assess these states systematically, welfare researchers highlight the importance of integrating various non-invasive, animal-based indicators of behavioral and physiological health [2,3]. For example, for decades, ethologists have monitored behavior by conducting observations on individual animals and then creating activity budgets [4]. More recently, welfare scientists have made a concerted effort to non-invasively measure physiological biomarkers that reflect an animal’s health and overall welfare status (e.g., glucocorticoids, immunoglobulin-A, dehydroepiandrosterone, and alpha-amylase) [3,5]. This paper examines how species-specific reference intervals can be created for both behavioral and physiological measures, offering a more fine-grained approach for determining whether individuals of particular age/sex classes fall within the expected range for the variable of interest. We specifically focused on chimpanzees (*Pan troglodytes*) in this study, calculating reference intervals for behavioral states and events, behavioral diversity, fecal glucocorticoid metabolites (GCM), and fecal immunoglobulin-A (IgA).

For decades, welfare scientists have created activity budgets to provide insight into how an individual spends their time [4]. Specifically, the researcher can calculate the proportion of time spent in particular behavioral states (e.g., inactive, play), as well as the rate of particular behavioral events (e.g., yawn, supplant), each of which is defined in a species-specific ethogram. By conducting observations across various seasons and reproductive states for different age/sex classes, baseline data can be compiled and used for future comparisons within populations or for intra-individual monitoring. However, activity budgets have limited value if we do not identify benchmarks or intervals that allow us to determine whether an individual is experiencing a good state of welfare. With behavioral data, it is also possible to calculate behavioral diversity. Behavioral diversity may serve as a positive welfare indicator, as individuals who live in stimulating environments and experience positive events are more likely to express a diverse repertoire of species-typical behavior [6,7,8,9]. Indeed, previous studies identified inverse relationships between behavioral diversity and more traditional welfare measures, including glucocorticoids [10,11,12] and stereotypic behaviors [7,8,9,13,14]. 

In terms of physiological biomarkers, animal welfare scientists most commonly focus on monitoring glucocorticoids (or glucocorticoid metabolites) to evaluate an individual’s physiological response to intrinsic and extrinsic stressors—some of which are considered beneficial and part of the animal’s natural history (e.g., courtship, hunting) [15,16,17]. Healthy hypothalamic-pituitary-adrenal (HPA) axis function allows individuals to maintain homeostasis and involves a negative feedback loop that suppresses the production of glucocorticoids after the removal of a stressor [15,18]. Animals exposed to repeated or chronic stressors may experience an upregulation of glucocorticoids, dysregulation of the HPA axis, and pathophysiological effects on metabolic function, cognition, mood, reproduction, and immune function [15,18,19,20,21,22,23]. Fortunately, welfare scientists can collect non-invasive fecal samples to regularly monitor long-term adrenocorticoid activity via glucocorticoid metabolites. After establishing an animal’s baseline hormone concentrations, it is possible to refer to these values to evaluate the focal individual’s adrenal response to various social, environmental, or husbandry changes. 

In recent years, immunoglobulin-A (IgA) has been used as a biomarker of physiological stress in studies of humans and laboratory animals. IgA, which can be measured via blood or non-invasively via mucosal surfaces, plays a critical role in mucosal immunity and is associated with physical health and psychological well-being [24,25], Reviewed by [5]. Exposure to chronic or repeated stressors, both physical and psychosocial, may suppress IgA concentrations, which can negatively impact immune function in humans [26,27,28,29,30]. Alternatively, there is evidence of acute increases in IgA after both humans and non-humans experience pleasant stimuli and positive emotional states [31,32]. Furthermore, when individuals experience good welfare, concentrations of IgA may stabilize at higher levels, lowering disease risk [24,33]. Indeed, there is evidence that higher levels of positive affect are associated with better current and future health outcomes in humans Reviewed by [31].

Recently, there has been a call to integrate IgA into welfare assessments for animals living in professional care. In dogs (*Canis familiaris*), salivary IgA was negatively correlated with salivary cortisol and also associated with behavioral assessments [34]. Specifically, while low levels of IgA were found in dogs “exhibiting stress”, dogs that were considered “calm” and “confident” had high levels of IgA [34]. For shelter cats (*Felis catus*), those who received a petting treatment (four times daily over a 10-day period) exhibited higher fecal IgA, were better able to maintain their mood, and were less likely to exhibit behaviors associated with frustration and anxiety [35,36]. Moreover, as compared to controls, cats that participated in positive reinforcement training had higher fecal IgA [37]. Although IgA and glucocorticoids were not correlated in a study of zoo-housed Asian elephants (*Elephas maximus*) see also [38,39], one subject who experienced a severe systemic illness exhibited notable increases in fecal IgA and glucocorticoids [40]. Finally, for zoo-housed Sichuan golden monkeys (*Rhinopithecus roxellana*), Huang and colleagues [41] identified a positive relationship between immunoreactive cortisol and fecal IgA, with the latter being significantly higher in the summer when microbial load and variety peak. Clearly, interpretations of IgA concentrations must be made with caution, as IgA may increase as the body prepares to combat pathogens and toxins, as well as in response to acute stressors such as restraint [5,42]. Overall, given how closely related chimpanzees are to humans, and the positive relationship between IgA and both physical health outcomes and positive emotional states in humans, we expect IgA to be a useful biomarker for chimpanzee welfare.

After collecting behavioral and physiological data, researchers should consider developing reference intervals for future comparisons. Reference intervals—created from large datasets—are ranges of values from physically healthy subjects and are utilized by veterinarians to assess the health of individual animals [43]. It is common for veterinarians to compare a subject’s values to clinicopathologic reference intervals established for not only that individual but also for healthy conspecifics [44,45]. Recently, species-specific reference intervals were developed for various physiological biomarkers for four cetacean species, including both blood variables (e.g., sodium, lymphocytes) and fecal hormone metabolites (e.g., cortisol, aldosterone, DHEA) [43,46]. Lauderdale and colleagues [47] suggest that “reference intervals and values provide an important comparative diagnostic reference tool set for future health assessments…” (p. 5). Indeed, animal care professionals can use these metrics to monitor hormone concentrations and determine when individuals fall outside of the expected range. Ultimately, researchers can use these data to make evidence-based decisions, gaining insight into how management practices and environmental conditions may impact welfare. 

We aimed to develop reference intervals—for both behavioral and physiological measures—for chimpanzees. There are approximately 1330 chimpanzees living in the United States, with roughly 240 of those being cared for in 30 zoological institutions accredited by the Association of Zoos and Aquariums (AZA). Chimpanzees are highly social and have complex behavioral needs. Those living in professional care often exhibit species-appropriate behaviors indicative of good welfare (e.g., social grooming, play) [48]. However, these individuals are also at risk of developing behaviors that may be indicative of poor welfare (e.g., regurgitation and reingestion, hair plucking, coprophagy), becoming highly aggressive, and failing to demonstrate appropriate social, breeding, and/or maternal behaviors [49,50]. Individuals with atypical early life experiences are especially prone to developing aberrant behavior and exhibiting social deficiencies [51,52,53]. In a 2016 study of 26 AZA-accredited facilities with chimpanzees, 64% of the sample population had engaged in abnormal behavior in the prior two years, and 48% engaged in abnormal behavior other than coprophagy [54]. Due to chimpanzees’ unique individual needs, preferences, and rearing histories, as well as welfare concerns from both animal care professionals and the public alike, we chose this species for a large-scale welfare study. 

Specifically, we partnered with 16 AZA-accredited facilities to collect data on 41 adult chimpanzees with no known health or welfare concerns. For these subjects, we conducted three 30-min focal observations per week and non-invasively collected daily fecal samples, which we analyzed for concentrations of fecal GCM and fecal IgA. We then created reference intervals for behavioral states, behavioral events, behavioral diversity, fecal GCM, and fecal IgA. We also determined whether these behavioral measures, fecal GCM, and fecal IgA, varied by age and/or sex. 

## 2. Materials and Methods

Subjects included 41 adult chimpanzees (18M and 23F) with no known health or welfare concerns. We included subjects from 16 zoological facilities accredited by AZA. To do so, each facility indicated how many chimpanzees could contribute to the study and provided a list of all individuals. After excluding subadults and individuals with known health or welfare concerns, subjects were chosen using a random number table. All subjects lived in mixed-sex groups composed of at least some individuals of breeding age, except for three subjects from one facility who were all geriatric. Adult chimpanzees were defined as individuals between 13 and 34 years of age, while geriatric chimpanzees were defined as individuals 35 years of age or older [55,56]. Our sample did not include pregnant or lactating females. Each subject contributed behavioral and physiological data for nine months (a 13-week baseline period followed by a 6-month WelfareTrak^®^ period, during which caretakers completed weekly welfare surveys using the WelfareTrak^®^ application, Chicago Zoological Society, Brookfield, IL, USA). For the current study, we specifically analyzed the data from the 13-week baseline period (see below). As described in more detail below, one subject was removed from the analysis for not having a sufficient number of fecal samples and observations.

The final dataset included 22 females and 18 males, ranging in age from 13 to 48 years at the start of data collection. Females had a mean age of 32.45 (±9.17 SD) years. Males had a mean age of 28.56 (±10.96 SD) years. Twenty-seven of the individuals were considered adults, and thirteen individuals were considered geriatric. The mean age of adult chimpanzees was 24.85 (±6.16 SD) years. The mean age of geriatric chimpanzees was 42.85 (±3.44 SD) years.

### 2.1. Behavioral Observations

Volunteers or staff from each facility filmed each subject for a 30-min focal follow observation three times per week between March 1, 2016 and November 30, 2016 (data collection started one month later for one subject and two months later for two other subjects, due to late recruitment into the project). We conducted continuous sampling of states (e.g., inactive, feed/forage) and all-occurrence sampling of events (e.g., affiliative touch, yawn). We instructed facilities to: (1) conduct observations at various times of the day (i.e., to balance morning and afternoon observations) to capture a comprehensive picture of the subject’s behavioral repertoire, and (2) do not observe the animals in their holding areas. For the current study, only the data collected during the first 13 weeks were included in the analysis. We did not analyze the full dataset, as facilities were encouraged to introduce changes to the husbandry routine and environment after Week 13, which may have impacted behavior. Videos were coded by staff and trained volunteers from the Chicago Zoological Society (CZS) using BORIS software (Torino, Italy), [57] according to a specified ethogram of species-appropriate behaviors adapted from Ross and Lukas [58]. All staff and volunteers reached r > 0.80 inter-rater reliability before coding videos. The behaviors analyzed for this study and their definitions are presented in Appendix A. 

### 2.2. Physiological Measures

We asked caregivers at each facility to collect the first defecation produced by the study animal each day for the duration of the study. To distinguish a study animal’s samples from its groupmates’, subjects received baking-grade food coloring (AmeriColorTM soft gel paste, Placentia, CA, USA). Samples were stored in a −20 °C freezer at each facility, then shipped overnight on dry ice to CZS and stored in a −20 °C freezer. 

Each fecal sample was weighed (0.5 g ± 0.05 g; exact weights were recorded into Excel) and placed into two separate 16 × 125 mm polypropylene tubes for fecal GCM and fecal IgA analysis using an analytical scale (Mettler balance, model #AB104-5). As a backup, approximately 5 g of leftover feces from each sample was placed into 12 × 75 mm polypropylene tubes. All tubes were stored at −20 °C until use.

#### 2.2.1. Fecal Glucocorticoid Metabolite (GCM) Measurement

Tubes for fecal GCM analysis were removed from the −20 °C freezer one day prior to hormone analyses and extracted using 5 mL of 80% ethanol in dH2O. Tubes were vortexed for approximately 30 s, and then placed overnight on a rotator set to 30 rotations per minute (Fisher Labline Maxi Rotator, model #4631, Fisher Scientific, Waltham, MA, USA). The morning of analysis, tubes were centrifuged at 1500 rpm for 15 min (Marathon 3000R centrifuge, model #120, Fisher Scientific, Waltham, MA, USA). One milliliter of supernatant from each sample was pipetted into a new 12 × 75 mm polypropylene tube containing 1 mL of assay buffer (0.1 M phosphate buffered saline containing 1% BSA, pH 7.0) to produce a 1:10 dilution. Immediately following the extraction steps, samples were assayed using a commercially available corticosterone EIA kit (Enzo Life Sciences, Ann Arbor, MI, USA, catalog #901-097). Plates were read on a spectrophotometer (Dynex MRX Revelation, Dynex, Chantilly, VA, USA) at an optical density of 405 nm. 

Numerous studies used these same techniques for fecal GCM analysis, e.g., [59,60,61,62], and a biological validation (i.e., ACTH challenge) for chimpanzees has been conducted in saliva [63] and feces [64]. Variability between assays (inter-assay CVs) was monitored using high and low controls across all plates. To determine intra-assay variability, a single sample was repeated 10 times on a single plate. 

Biochemical validation of both fecal GCM and fecal IgA assays consisted of a linearity test to determine parallelism with the standard curve, in addition to a recovery test to measure the concentration of exogenous analytes. To establish parallelism, serial two-fold dilutions of a sample pool were tested for potential interference in the sample matrix, linearity with the standard curve, and the appropriate dilution factor at which to run the samples. The optimal sample dilution for fecal GCM was 1:500, as this dilution was closest to 50% binding of the sample pool, and 1:75 for fecal IgA. 

Recovery of exogenous fecal GCM was measured by spiking one diluted sample with each of the 5 highest standards in separate tubes. Each standard contained a known amount of hormone ranging from 250–4000 pg/mL. The average percent recovery was calculated by dividing the measured concentration of fecal GCM by the expected concentration of fecal GCM, multiplied by 100.

The cross-reactivity of the Enzo Life Sciences corticosterone antibody is 100% corticosterone, 28.6% desoxycorticosterone, 1.7% progesterone, 0.28% tetrahydrocorticosterone, 0.18% aldosterone, 0.13% testosterone, and any other steroids <0.05%. Assay sensitivity was 26.99 pg/mL, and the intra-assay coefficient of variation was 3.94% at 80.48% binding with an average concentration of 38.76 pg/mL (n = 10). Inter-assay variation was determined using a high and low control, 10.05% CV at 28.08% binding and 19.15% CV at 49.86% binding, respectively. The average recovery of exogenous corticosterone was 103.34% (SD = 24.50). We expressed all fecal GCM concentrations as ng/g of wet feces. 

#### 2.2.2. Fecal Immunoglobulin-A (IgA) Measurement

We followed the methods of Lantz et al. [65] to extract and assay fecal immunoglobulin-A. Briefly, IgA was extracted using 5 mL of 1× phosphate buffered solution (PBS; 5.42 g NaH2PO4, 8.66 g Na2HPO4, 8.7 g NaCl, 0.8 g NaOH added to 1 L dH2O, pH adjusted to 7.2 using 5M NaOH). Tubes were vortexed for 30 s, placed on the rotator (Fisher Labline Maxi Rotator, model #4631, Fisher Scientific, Waltham, MA, USA) for 2 h, and then centrifuged at 1500 rpm for 15 min. One milliliter of supernatant was then transferred to a new 1.7 mL polypropylene Eppendorf tube and stored in a −20 °C freezer until analysis. We assayed the samples using a commercially available IgA-human ELISA (Bethyl Laboratories, Montgomery, TX, USA, catalog #E80-102). Plates were read using a spectrophotometer (Dynex MRX Revelation, Dynex, Chantilly, VA, USA) at an optical density of 405 nm. 

The Bethyl Laboratories IgA-human antibody is 100% specific to human IgA, and no further testing on cross-reactivity with other species has been studied at this time. There is no cross-reactivity with other human immunoglobulins or serum proteins. The assay range is 7.8–500 ng/mL. The intra-assay coefficient of variation was 4.23% at 17.76% binding with an average concentration of 69.98 ng/mL (n = 10). Inter-assay variation was determined using a low and high control, 17.96% CV at 79.38% binding and 13.15% CV at 23.45% binding, respectively. We expressed all IgA concentrations in µg/g of wet feces.

### 2.3. Statistical Analysis

Analyses were completed using the software package IBM SPSS Statistics 27. All fecal GCM and fecal IgA values >3 SD were considered outliers and removed. Furthermore, subjects missing >50% of their behavioral observations, fecal GCM values, or fecal IgA values were removed, which resulted in one subject being removed from the analysis. 

As fecal samples were assumed to contain GCM and IgA concentrations representative of the 24-h period prior to the morning they were collected, all data from fecal samples were shifted to the prior day. Means were calculated for both fecal GCM and fecal IgA for the entire 13-week baseline period. 

For the behavioral data, the time visible was corrected for each observation. Behavioral states were converted to proportions by dividing the number of seconds spent performing the behavior of interest by the total minutes visible for that observation. Behavioral events were converted to rates by dividing the number of occurrences of the behavior of interest by the total minutes visible for that observation. Next, we calculated means for each behavior for the entire baseline period. 

Behavioral diversity was calculated using the Shannon Diversity Index [66]. Behavioral diversity is notated as H, with higher values signifying a greater number of behaviors and/or a more even distribution of behaviors. The Shannon Diversity Index (H) is calculated as
(1)$$H=-\overset{s}{\underset{i=1}{\sum }}p_{i}\ln p_{i}$$
where *p_i_* is the proportion of the behavior category. Behaviors that were included in the Shannon Diversity Index calculation were: contact; feed/forage; groom, self-directed; groom, social agent; groom, mutual/multiple; locomotion, horizontal; locomotion, vertical; object manipulation, enrichment; object manipulation, other; play, social; play, solitary.

For the reference intervals, linear regressions were used to compare the physiological measures, behavioral diversity, behavioral states, and behavioral events by sex, age, and age squared (age^2^). The age^2^ variable was calculated by squaring the age variable. The significance level for the physiological measures was defined as *p* < 0.05, and the significance level for the behavioral measures was defined as *p* < 0.05. The data were randomly split 70/30 to create a 70% training set and a 30% testing set. The data were split by observation to include individuals from multiple locations with a variety of sexes and ages in both sets. Prediction equations were extracted from the linear regression models using the 70% training data set. The predicted values were compared to observed values by assessing the output from the models, means, and standard deviations of the predicted values relative to the observed values. If significant predictors were identified by the linear regressions, models were run with only the significant predictor/s. The final models were developed using only the respective significant predictors. If no significant predictors were identified, the reference intervals were established from values that were within two standard deviations from the mean.

## 3. Results

Prediction equations from training and testing data sets are given in Table 1. Mutual/multiple social grooming significantly decreased as individuals aged. There was a positive effect of age^2^, suggesting that the effect increased with age; that is, as chimpanzees aged, the time spent engaging in mutual/multiple grooming decreased faster. Sex was a significant predictor in three models. When compared to females, males spent a significantly higher proportion of time performing aggressive contact behaviors and displaying to other chimpanzees. Males performed sexual examination behaviors at a higher rate than females. Table 2 displays the results of the linear regression models. General reference intervals (not accounting for specific age for mutual/multiple social grooming) are presented in Table 3. For the variable in which age and age^2^ were significant predictors, the reference intervals in Table 3 display age classes rather than specific ages. 

## 4. Discussion

This is the first known study to create species-specific reference intervals for chimpanzees living in professional care. We analyzed data from 22 female and 18 male chimpanzees, ranging in age from 13 to 48 years, to calculate reference intervals for behavioral states, behavioral events, behavioral diversity, fecal GCM, and fecal IgA. Reference intervals were developed by collecting data on physically healthy subjects with no welfare concerns and calculating ranges of values for the variables of interest [43]. The reference intervals displayed in Table 3 were calculated using data from both males and females, as well as adult and geriatric individuals, unless otherwise noted. Specifically, the reference intervals for mutual/multiple social grooming are presented separately for adults and geriatric individuals, as this measure varies by age. In addition, the reference intervals for performing contact aggression, displays to other chimps, and sexual examination behaviors are presented separately for males and females, as these behaviors vary by sex. Behavioral diversity did not vary by age or sex. 

As noted above, we discovered either sex or age differences for a handful of behaviors. First, we found that mutual/multiple social grooming significantly decreased with age and that this effect was greater for older individuals. Previous studies reported that adult chimpanzees engage in mutual grooming to signal their willingness to continue a grooming bout (i.e., the immediate investment hypothesis) and to build/maintain social bonds, e.g., [67,68]. Unfortunately, no large-scale, systematic studies have examined how mutual grooming changes throughout adulthood. We postulate that as chimpanzees age, their role in the social hierarchy becomes more solidified [69,70,71,72], and as such, grooming between partners may become more uni-directional (favoring the higher-ranked individual) instead of bi-directional [73,74,75].

We also discovered that, when compared to females, males spent a higher proportion of time performing aggressive contact behaviors and displaying to other chimps. This is consistent with previous studies, which reported that males exhibit physical aggression up to 14 times more often than females [76,77,78,79]. In fact, males “…rely on frequent dominance displays and aggressive contact with conspecifics to navigate within-group dominance hierarchies” [77], p. 2; [80]. Furthermore, de Waal [81] noted that females do not typically perform bluff displays, and if they do, the displays are incomplete or shortened. Similarly, our finding that males are more likely than females to perform sexual examination behaviors is expected, given that males inspect the anogenital swellings of females to assess reproductive condition [82,83,84].

We did not find sex or age differences for fecal IgA. The lack of sex differences for IgA is consistent with findings from previous animal studies, e.g., rats [85], Sichuan golden monkeys [41], and including a study of wild chimpanzees. Lantz and colleagues [65] discovered that fecal IgA concentrations did not significantly differ between males and females or females in different reproductive states. These authors also reported no age differences, though there was a trend for mature chimpanzees (older than 10 years of age) to have higher mean IgA concentrations than immature individuals. Our study did not include immature individuals but did differentiate between adults (13–34 years of age) and geriatric individuals (>35 years of age). 

Similarly, fecal GCM did not vary by sex or age. In terms of age, our results are consistent with the findings of previous studies that examined salivary cortisol for 24 zoo-housed chimpanzees ranging in age from 8 to 50 years [86] and hair cortisol for 25 professionally managed chimpanzees ranging in age from 11 to 45 [87,88]. However, a study of wild female chimpanzees (aged 10–55 at the start of the study) reported that both mean age and relative age were significant predictors of cortisol [89]. Specifically, older females had higher cortisol levels than younger females, and individual females exhibited increasing cortisol levels over time. In terms of sex differences, the results from previous studies are mixed. For instance, while Yamanashi and colleagues [87] did not find sex differences for chimps in professional care, other studies found males exhibited higher mean hair cortisol levels than females [88,90]. Inconsistent findings may be due to several factors, such as differences in group composition (e.g., sex ratio), reproductive status (e.g., the presence of lactating or pregnant females), and stability of the dominance hierarchy. Our study, which included adult and geriatric chimpanzees living in mixed-sex groups with no lactating/pregnant females, contributes to the growing body of biomarker data for this species. 

Ultimately, the reference intervals presented here form a diagnostic tool that can be integrated into future health and welfare assessments. Facilities can utilize these metrics to monitor individual animals and “flag” when values for behavioral or physiological measures fall outside of the expected range. While it would be preferable to collect long-term data (spanning various seasons and reproductive states) for all chimpanzees, animal care professionals may only have a handful of samples or a limited amount of behavioral data available for a particular individual. In these cases, the mean for the variable of interest can be compared to the appropriate reference interval. For example, animal care professionals or scientists can analyze a few weeks of fecal samples to calculate the mean fecal GCM and compare that value to the range presented here. If the value falls outside of the range, the animal care team can discuss next steps for investigating this concern. For instance, the team may decide to collect additional data (e.g., more fecal GCM data, behavioral data, etc.) or introduce a change to the environment/husbandry routine in an attempt to “move” the value into the expected range. Though the application of the information to management changes may occur weeks after physiological samples are collected, it is beneficial to have a variety of supportive evidence, and these reference intervals are another tool in the welfare assessment toolkit. It is crucial to remember that these reference intervals were created using data from healthy adults with no known welfare issues, so one might expect animals with documented welfare concerns to fall outside of the ranges presented here. Furthermore, researchers must keep in mind that these values were calculated using data from adult and geriatric individuals and may not be appropriate for younger individuals, pregnant females, or males living in bachelor groups. 

## 5. Conclusions

This is the first study to create reference intervals for chimpanzees for both behavioral and physiological measures. Specifically, we presented ranges for behavioral states, behavioral events, behavioral diversity, fecal GCM, and fecal IgA for adult males and females living in professionally managed care. Furthermore, we discovered that geriatric subjects spent less time engaged in mutual/multiple grooming than younger adults and that males performed more contact aggression, displays to other chimps, and sexual examination behaviors than females. In the future, welfare scientists, veterinarians, and animal care professionals will have the ability to compare a chimpanzee’s values using the intervals presented here. Ultimately, animal care professionals can use these data to make evidence-based decisions, gaining insight into how management practices and environmental conditions impact animal welfare. 

## Figures and Tables

**Table 1 animals-13-00639-t001:** Fit of predictions on the training and testing datasets from the linear regression models. Training sets included 70% of the data, and testing sets included 30% of the data.

Variable	Data Set	n *	Equation	r^2^
Aggressive, Contact, Agent	Training	28	0.005 + −0.005 × Sex	0.431
	Testing	12	0.018 + −0.016 × Sex	0.699
Display, Chimps	Training	28	0.072 + −0.067 × Sex	0.457
	Testing	12	0.125 + −0.099 × Sex	0.642
Groom, Social, Mutual/Multiple	Training	28	24.568 + −1.440 × Age + 0.022 × Age^2^	0.536
	Testing	12	1.417 + 0.332 × Age + −0.007 × Age^2^	0.356
Sex Exam, Agent	Training	28	0.003 + −0.002 × Sex	0.469
	Testing	12	0.012 + −0.012 × Sex	0.716

* n = number of subjects, sex: male = 0, female = 1.

**Table 2 animals-13-00639-t002:** Model estimates for predicting behaviors with significant variables. Linear regression models (training dataset) included the explanatory variables found to be significant predictors. Possible fixed factors included: sex, age, and age^2^.

Model	Variable	Parameter	Estimate	Standard Error	*t*	*p*
i	Aggressive, Contact, Agent	Intercept	0.005	0.002	3.122	0.004
		Sex	−0.005	0.002	−2.433	0.022
ii	Display, Chimps	Intercept	0.072	0.020	3.626	0.001
		Sex	−0.067	0.025	−2.617	0.015
iii	Groom, Social, Mutual/Multiple	Intercept	24.568	6.800	3.613	0.001
		Age	−1.440	0.475	−3.029	0.006
		Age^2^	0.022	0.008	2.854	0.009
iv	Sex Exam, Agent	Intercept	0.003	0.001	4.368	<0.001
		Sex	−0.002	0.001	−2.708	0.012

**Table 3 animals-13-00639-t003:** General reference intervals for physiological measures, behavioral diversity, and behaviors. General 95% reference intervals are given below.

Variable	Age Range	n *	Reference Interval
Physiological Measures	(Years)		
Fecal GCM	13–48	40	27.21–587.93 ng/g
Fecal IgA	13–48	40	11.43–45.94 µg/g
Indices	(Years)		
Behavioral Diversity	13–48	40	1.22–2.00
Event Behaviors	(Years)		(Behaviors per minute)
Affiliative Touch, Agent	13–48	40	0.00–0.02
Copulation	13–48	40	0.00–0.01
Sexual-Exam, Agent (Male)	13–48	18	0.00–0.21
Sexual-Exam, Agent (Female)	16–45	22	0.00–0.03
Sexual-Present, Agent	13–48	40	0.00–0.00
Supplant, Agent	13–48	40	0.00–0.01
Yawn	13–48	40	0.00–0.08
State Behaviors	(Years)		(%)
Aggressive, Contact, Agent (Male)	13–48	18	0.00–0.03
Aggressive, Contact, Agent (Female)	16–45	22	0.00–0.01
Aggressive, Non-Contact, Agent	13–48	40	0.00–0.08
Contact	13–48	40	0.07–12.24
Display, Chimps (Male)	13–48	18	0.00–0.31
Display, Chimps (Female)	16–45	22	0.00–0.06
Display, Humans	13–48	40	0.00–0.14
Feed/Forage	13–48	40	3.13–27.15
Groom, Social, Mutual/Multiple (Adults)	13–34	27	0.00–15.31
Groom, Social, Mutual/Multiple (Geriatric)	37–48	13	0.02–9.58
Groom, Self-Directed	13–48	40	1.25–13.89
Groom, Social, Agent	13–48	40	0.03–9.49
Human Interaction, Orientation, Public	13–48	40	0.00–2.73
Human Interaction, Orientation, Staff	13–48	40	0.00–2.94
Inactive	13–48	40	29.28–69.00
Locomotion, Horizontal	13–48	40	2.94–15.90
Locomotion, Vertical	13–48	40	0.20–3.71
Masturbation	13–48	40	0.00–0.00
Object Manipulation, Enrichment	13–48	40	0.00–2.53
Object Manipulation, Prepared Enrichment	13–48	40	0.00–6.22
Object Manipulation, Other	13–48	40	0.17–5.82
Play, Social	13–48	40	0.00–3.76
Play, Solitary	13–48	40	0.00–0.22
Scratch, Gentle	13–48	40	0.10–1.03
Scratch, Rough	13–48	40	0.07–1.40
Submission, Agent	13–48	40	0.00–0.03

* n = number of subjects.

## Data Availability

The data presented in this study are available in the article and Appendix A.

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
