# Peer review of "Integrating Reference Intervals into Chimpanzee Welfare Research"

_animals, 2023, doi:10.3390/ani13040639_

Round 1

Reviewer 1 Report

The Authors present an original article called “Integrating Reference Intervals into Chimpanzee Welfare Research”. I believe this topic to be extremely important and am delighted and grateful to see colleagues dedicating their time and energy to provide these results. I definitely recommend this article to be published, although suggest some modifications for this special issue and some recommendations in case they foresee to keep working on this topic for future publications.

From the abstract I expected an article more focused on behavior or at least equally focused on physiological biomarkers and behavior. However, the project seems to be strongly leaning towards the biomarkers. As it is, this article is already impressive and extremely valuable, yet I think it would have been great if more resources would have been dedicated to the behavioral part. That being said, my expertise lies within the behavioral section and although I did not spot any mayor problems in the sections regarding biomarkers, I do lack the experience to properly evaluate those parts.

Following please find some suggestions and recommendations:

Introduction:

The authors provide many examples and information regarding physiological biomarkers, most of all immunoglobulin-A (IgA), yet provide quite little information on behavioral indicators and how they could indicate poor or good wellbeing. Basically, the authors only mention the compilation of activity budgets and the possibility to calculate behavioral diversity. I would recommend adding some more information of behavioral indicators of poor or high levels of well being based on the chimpanzee’s behaviors. Towards the end, some references to abnormal behavior can be found, yet abnormal behavior has not been observed, recorded or analyzed.

48: “activity budgets are being created”. This is true, but it might be interesting here to state, that we miss intervals to be able to evaluate their behavior to be either higher, lower or within the expected interval of healthy and well cared for chimpanzees. You could highlight here that simply compiling an activity budget is not enough to evaluate their state, if we lack guidelines or data sets to compare these budgets with.

48-50: You mention the use of physiological biomarkers here for the first time. I suggest to name a few of the more typical ones, already in this paragraph.

62-63: baseline data can be compiled and used for future comparison. True but I would specify that this only allows the comparison within the population and to one self over time. Yet studies like this one presented allow to make comparison with other captive populations.

118-119: the authors explain that reference intervals are being established based on data from healthy animals, yet healthy is not necessarily equivalent for high quality of life. Have they been evaluated in any way and identified as individuals to have high levels of well being?

147: “collected 3 30min focal observations per week.” This needs to be in the method section and requires more detailed explanations

Methods:

155: 41 adult chimpanzees (18M, 23F) but in the abstract the authors say its 40 adult chimpanzees (22 females, 18 males). In line 166 the authors satte that one individual was removed from the analysis, which might explain the difference, yet I find it quite confusing to see different numbers in different sections. Also please explain why that individual was removed from analysis.

156: “recruited” does not seem the correct term here.

168: “2.1 Behavioral observations”:

-          How were the video recordings distributed throughout the week and the hours of the day? Always at the same time and equal conditions (for example, never at feeding times, always in indoor or always in outdoor areas?) Such conditions should be controlled

-          172. The authors state: “For the current study, only the data collected during the first 13 weeks were included in the analysis.” Which is not a problem in general, but if you provide this information you should also explain why you choose to only use the first 13 weeks.

-          I suggest to provide here more specific information regarding the data collection and methodology. With the information at hand you can only guess if they use interval sampling + all-occurrence or continuous sampling + all occurrence. Thus, I am not criticizing the methodology itself, but rather ask to explain in more detail the methods used to collect behavioral data. Although from some explanations in the following chapters I understand that continuous observations were used in combination with all occurrence observations.

181: If more detailed information was provided regarding the methodology than I suggest to move the Ethogram (Table1) into the supplementary material and only provide here a brief overview stating that some behaviors were recorded es states and others as Events due to ….

264: eliminate the word “during”

270: Please add a citation to the Shannon Diversity Index

3. Results

296-302: This paragraph talking about the sample population´s age should be moved to the method section

323: Table 4 contains an interval entry for “sleep”. However, I did not see sleep in your ethogram (Table1), and generally would suggest to avoid such an item in a behavioral ethogram as an observer can hardly tell objectively if an animal is sleeping or simply inactive with their eyes closed. This needs to be properly defined in the ethogram, taken out or perhaps fused together with “inactivity”.

323: Table 4 – Some behavioral Intervals seem extremely broad

4. Discussion

Generally, the discussion is fine, however provides very little discussion and argumentation regarding the findings. It reads more like a more reader friendly version of the result section plus a short conclusion. I would suggest to elaborate a bit more, arguing the usefulness of these results and go a bit into detail regarding the established intervals.

General suggestions:

I suggest to more precisely explain how individuals were selected, as you state only individuals in good health were selected. But did they also account for high levels of well being? Health and well being do not always go hand in hand and looking at the established intervals I was surprised to see some extreme values I would not necessarily expect to see in individuals characterized to exhibit high levels of well being.

You took sex and age into account, but what about other factors such as type/size of habitat and care management. What about group size? If you compare social behavior occurrences it might be important to take the group size into account. It might be too late for this publication, but perhaps should be mentioned as a potential influencing factor explaining big ranges in the intervals. Furthermore, this could be integrated in follow up projects.

I was surprised to see that “abnormal”, “stereotypical” or “excessive self-directed” behaviors are not part of the ethogram and were not analyzed. Does this mean that the study sample was established by excluding chimpanzees who are exhibiting any of these undesired behaviors? Or was this simply not recorded?

As stated, before I believe this article to be very valuable and think more research like this needs to be worked on and published. That being said, many of the intervals in the behavior state section show extremely wide ranges. Such as Contact (0.07-12.24), Feed/Forage (3.13-27.15), Groom adult (0.00-15.31) groom self (1.25-13.89), inactivity (29.27 -69.00) etc.. I believe this article is a crucial first step, but these intervals need to be narrowed down to achieve the desired reference interval catalogue. As an example, a chimpanzee not grooming any other chimpanzees, would still be within the suggested interval range; so would a chimpanzee dedicating 2/3 of his day being inactive or a chimpanzee spending 13% of his day grooming himself (what very likely would be considered over-grooming in many other studies).

I want to congratulate the authors to this article and am sure they will be able to make use of some suggestions to further improve this draft. Other suggestions might come to late to be included in this article, yet should not be a reason to not publish the paper as long as the authors mention them in the discussion. I sincerely hope that more research is being focused on such important topics.

Reviewer 2 Report

Thank you for the opportunity to read your manuscript. The study is clearly designed and presented, and the results of the study provide important and useful indicators of chimpanzee welfare that will be useful to zoo and sanctuary staff. I found the separation of the sampled population into adult and geriatric individuals particularly useful, and I think sanctuary staff will appreciate that consideration, as I imagine their chimpanzee population skews older than is true of zoos. The results are clearly presented such that future researchers can use this as a foundation for further refinement of these welfare indicators. I recommend publication, and I did not see anything needing revision.

The co-authors provide reference intervals for well-being in chimpanzees using behavioral (behaviors and behavioral diversity) and physiological (fecal glucocorticoid and immunoglobin-A) measures. The results will be useful for those caring for chimpanzees in sanctuary and zoo settings, in particular because of the variety of measures. The measures were applied to a large (40 adults) and diverse (age, gender) chimpanzee population, which speaks to the reliability of the measures and their applicability for other chimpanzee individuals.

The topic is original and relevant, and it addresses a gap in the field. To date, it has proved challenging to assess welfare of captive primates, particularly because individuals in captivity vary widely in gender, age, and rearing history. Additionally, some facilities may lack resources to collect physiological measures of well-being, so the diversity of the measures provided in the article provides options that can be applied when resources are scarce. The measures are created for chimpanzees, but I imagine that rather quickly the study will be replicated/extended to other large-bodied apes and other primates.

The study is unique in bringing together four indicators of welfare (two behavioral and two physiological measures). For behaviors, the authors use a standardized ethogram that is widely available and often used in the study of chimpanzees. The physiological measures can be noninvasively collected from fecal samples. Their description of four measures enables facility caregivers to choose which one(s) will be most effective to apply in their setting (sanctuary or zoo).

I found the manuscript to be clearly, concisely, and logically written with good flow. The methodology is fully described. The study population includes 40 chimpanzees, which is a large sample size in primatological research. The population includes 18 males and 23 females, and the ages are varied (but all adult). This latter point is important in that captive chimpanzee populations skew older, and some specific well-being concerns may occur at later life stages. I have no changes to suggest.

The conclusions are consistent with the evidence and the arguments presented, and they do address the main question posed. The authors set out to produce reference intervals for captive chimpanzee welfare, and they do so for four measures in table 4. They emphasize the features of the reference population: adult, healthy, and both sexes. In future and if possible, it may be helpful to include younger individuals in the study population, particularly because in Africa, sanctuaries often receive immature chimpanzees whose well-being should be monitored.

I expect that Table 4 will be widely used at captive facilities. Table 2 is helpful in showing different measures for males and females for some behaviors (e.g., aggressive contact, display). The tables are well formatted and easily understood. The incorporation of the ethogram in the article (Table 1) increases the utility of the article for facility staff. 

Author Response

Reviewer 2 did not suggest any revisions, and we are appreciative of the highly positive feedback. We thank Reviewer 2 for taking the time and effort to review our paper. Please see responses to Reviewer 1.